# Technical Note: Spectral correction for cavity ringdown isotope analysis of plant and soil waters

Gabriel J. Bowen[1], Sagarika Banerjee[1], Suvankar Chakraborty[1]

[1]Department of Geology & Geophysics, University of Utah, Salt Lake City, Utah, 84112, USA

*Correspondence to*: Gabriel J. Bowen (gabe.bowen@utah.edu)

**Abstract.** The development of laser spectroscopic analysers has revolutionized isotope hydrology, dramatically increasing accessibility and reducing the cost of sample analysis. Despite their substantial benefits, these instruments are known to suffer from spectral interferences caused by small organic molecules that can bias measurements of some samples. Previous research has characterized this problem and tested a range of solutions for eliminating, detecting, or correcting influence in experimental
or natural samples, yet interlaboratory comparisons show that affected data are still being reported. Here, we use paired spectroscopic (Picarro L2130-i; CRDS) and mass spectrometric (IRMS) data from a diverse suite of soil and plant xylem water samples to characterize spectral interference effects on CRDS $\delta^2H$ and $\delta^{18}O$ data. Interference is minimal for soil water but widespread in plant samples, with 13% and 54% of samples exhibiting biases larger than 8‰ for $\delta^2H$ and 1‰ for $\delta^{18}O$, respectively. We develop multivariate statistical models that use analyser-reported spectral features to correct for interference.
These models account for 57% of the observed $\delta^2H$ bias and 99% of the $\delta^{18}O$ bias, and after correction the standard deviation of the CRDS-IRMS differences for plant samples (4.1‰ for $\delta^2H$ and 0.4‰ for $\delta^{18}O$) was similar to that for soil samples. Applying the models to CRDS measurements of water extracted from 1176 plants and 693 soils collected across diverse ecosystems improves the correspondence between plant and source soil water values and shows strong taxonomic differences in the prevalence of spectral interference. Our results show that spectral interference remains a significant concern in
ecohydrology, particularly for plant water extracted from many woody species. The success of our spectral correction models across a wide range of taxa and data generated from two different CRDS analysers suggests that *post-hoc* correction of these data may be a viable solution to the problem.

## 1 Introduction

The development of commercial laser spectroscopy instruments for the measurement of H and O isotopes in water has
revolutionized the fields of isotope hydrology and ecohydrology by dramatically reducing the cost and increasing the accessibility of analyses (Lis et al., 2007; Berman et al., 2009; Gupta et al., 2009; Chesson et al., 2010; Munksgaard et al., 2011). Despite the many advantages of these instruments, it was recognized early in their history that they may be susceptible to analytical bias for samples that contain compounds, particularly low-weight organic molecules, with spectral absorption features that overlap those of the water isotopologues (Brand et al., 2009). This susceptibility is of particular concern in

ecohydrological research, in which water is commonly extracted from soils and plant tissues, which may contain and contribute volatile organic compounds to the extracted sample. The potential impact of these impurities on laser-based isotope analyses has been documented extensively and has (understandably) contributed to scepticism of laser isotope analysis in ecohydrology and adjacent fields (West et al., 2010).

Since its recognition, the spectral interference problem has received substantial attention. This work has led to the proposal of three types of workarounds that attempt to either eliminate interfering compounds prior to analysis, identify and cull affected measurements, or correct for interference during *post-hoc* data processing. Solutions involving elimination of interference include off-line chemical purification procedures (West et al., 2010; Chang et al., 2016) and in-line combustion devices that covert organics to water and $CO_2$ (Martín-Gómez et al., 2015; Cui et al., 2021). Although these methods have been shown to
be effective in some cases, they increase the complexity and/or labour involved in the sample preparation workflow and have limitations in terms of the types and/or concentrations of compounds that they can effectively remove. Solutions in which contaminated samples are flagged and culled from datasets will also largely be instrument system-specific but may be more transferrable and represent a conservative approach to quality control. These include commercial software (West et al., 2011) and screening procedures developed in individual studies (e.g., Schultz et al., 2011; Lazarus et al., 2016). Although these
approaches are often successful in identifying contaminated samples testing shows that they are not always effective (West et al., 2011), and they may involve substantial data loss if contamination is prevalent. Solutions involving *post-hoc* data correction would be ideal in terms of maximizing the value of data without adding analytical overhead. Although effective correction algorithms have been published, these are of limited utility because most are unique to the specific interfering compound(s), instrument type, or even individual instrument involved (Hendry et al., 2011; Schultz et al., 2011; Schmidt et al., 2012; Lazarus
et al., 2016; Johnson et al., 2017). Recent work by Herbstritt et al. (Herbstritt et al., 2024) developed correction equations for current-generation CRDS analysers and suggested that correction based on reported $CH_4$ concentrations in the analyser cavity might be broadly useful but would need to be calibrated independently for different sample types and analysers. This approach has yet to be tested at scale.

Here, we report analyses of cryogenically extracted water from nearly 1,200 plant water and 700 soil water samples made using current-generation CRDS instruments. We benchmark these data against IRMS analyses of a subset of samples and observe frequent, but not ubiquitous, bias in the CRDS data for plant waters. We develop multivariate models that describe $\delta^2H$ and $\delta^{18}O$ bias as a function of instrument-reported spectral features and show that these models successfully correct bias in $\delta^2H$ and $\delta^{18}O$ values for waters analysed on two different CRDS analysers. Finally, we apply the models to the full dataset
to investigate the prevalence of spectral bias and assess the ability of the models to correct for bias across a large and diverse ecohydrological dataset.

## 2 Methods

Plant and soil samples were collected at 12 U.S. National Ecological Observatory Network (NEON) sites during the 2020 and 2021 growing seasons. Plant samples consisted of suberized stems, stripped of bark and sectioned, or shallow roots of non-woody species (grasses). Soil samples were collected from the mineral soil with a hand auger at up to 5 different depths below the soil surface, extending as deep as 95 cm at some sites. Samples were collected by NEON staff, stored in sealed 20 ml glass vials at room temperature, and returned to the Stable Isotope Facility for Environmental Research (SIRFER) at the University of Utah. SIRFER staff conducted cryogenic vacuum distillation of all samples using the methods of West et al. (2006).

All extracted samples were analysed for hydrogen ($\delta^2$H) and oxygen ($\delta^{18}$O) isotope values via CRDS. Two different Picarro L2130-i analysers were used (serial numbers HIDS2046 and HIDS2052), with an approximately equal number of samples analysed on each instrument. Plant samples were pretreated with activated charcoal for 48 hours. An inline combustion device was not used during analysis. The analytical setup and data reduction strategy were as described in Good et al. (2014), and all data processing was conducted using the CRDSutils R-package (Bowen and Blevins, 2024). Two laboratory reference waters were used for calibration (PZ: $\delta^2$H = 18.1‰, $\delta^{18}$O = 1.93‰; UT2: $\delta^2$H = -119.1‰, $\delta^{18}$O = -15.84‰ relative to the VSMOW2-SLAP scale) and a third (EV: $\delta^2$H = -72.3‰, $\delta^{18}$O = -10.16‰) was analysed repeatedly in each run to monitor drift and as a quality control. Analytical precision based on the analyses of EV across all analytical batches was approximately 0.3‰ for $\delta^2$H and < 0.1‰ for $\delta^{18}$O (1$\sigma$). Raw data files for all runs were screened using Picarro's ChemCorrect$^{TM}$ software.

A subset of 58 plant and 16 soil samples were also analysed by conventional isotope ratio mass spectrometer (IRMS). $\delta^{18}$O values were determined by $CO_2$ equilibration followed by chromatographic separation using a ThermoFisher GasBench II coupled with a MAT253 IRMS. $\delta^2$H values were determined by pyrolysis using a ThermoFisher TC/EA coupled with a Delta Plus IRMS. Data were calibrated against two reference waters (ZE: $\delta^2$H = -0.2‰, $\delta^{18}$O = -0.2‰; DI: $\delta^2$H = -123.0‰, $\delta^{18}$O = -16.5‰) and EV was analysed as a quality control material. Analytical precision (calculated as described above) was approximately 1.5‰ for $\delta^2$H and 0.15‰ for $\delta^{18}$O. IRMS analyses were conducted on residual water from the CRDS analysis vials. The water was transferred to sealed vials and stored for up to 9 months prior to IRMS analyses, creating the potential for evaporative fractionation from imperfectly sealed vials. IRMS data from 5 samples (4 plant, 1 soil) showed much lower (> 35‰) deuterium excess values ($d = \delta^2$H $-$ 8 x $\delta^{18}$O) than the CRDS analyses, suggesting evaporation during storage. These data, along with those from one plant sample with an anomalously high $d$ value (+29.4‰), were excluded from further analysis.

We compared CRDS and IRMS results directly and assumed that the IRMS data represented the true sample values. We used an iterative linear model selection process (regsubsets function; Lumley, 2024) to optimize models describing the $\delta^2$H and $\delta^{18}$O bias of CRDS measurements (i.e., $\delta_{CRDS} - \delta_{IRMS}$) as a function of 5 metrics reported in the CRDS output files that reflect

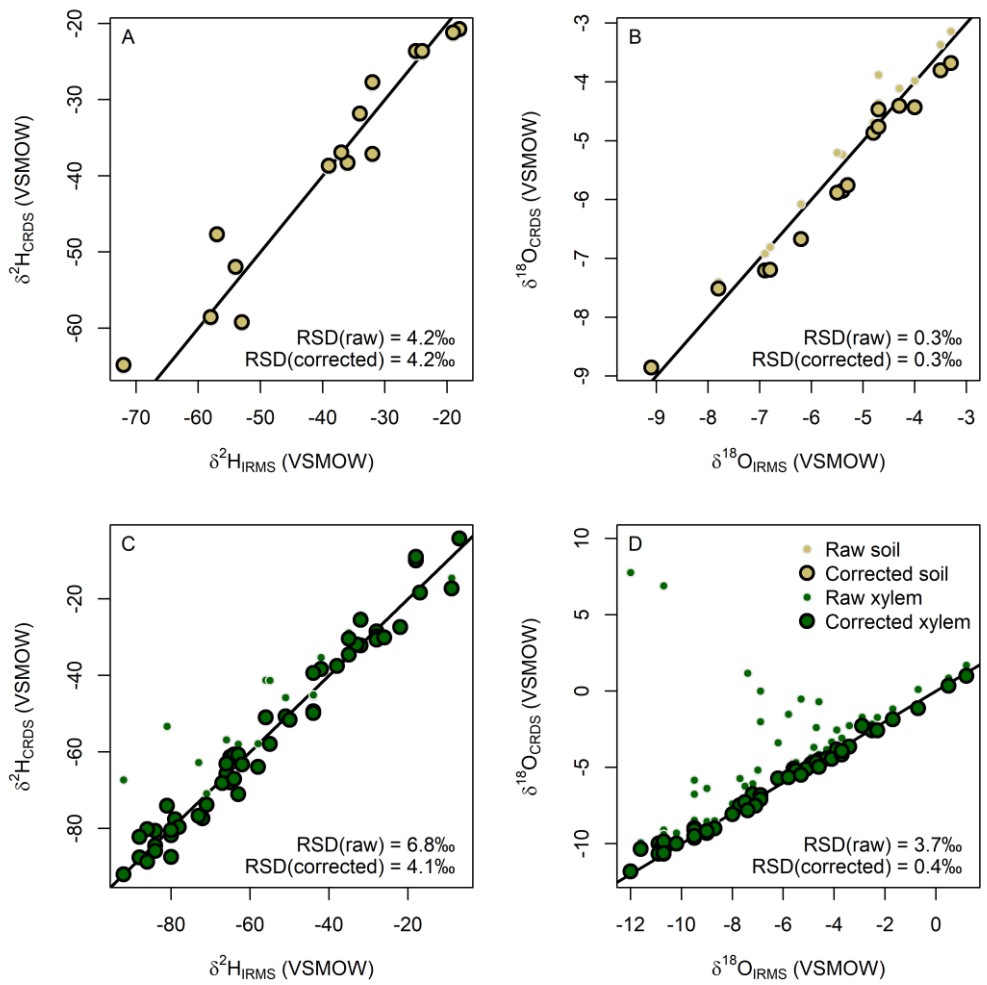

**Figure 1: Comparison of paired CRDS and IRMS measurements for soil (A and B) and plant xylem (C and D) water samples before and after correction of the CRDS data for spectral interference. Note that symbols showing the raw values of samples with little or no correction are obscured by the corrected data (e.g., all raw data in panel A). The black line in each panel shows the 1:1 relationship. RSD: residual standard deviation.**

the potential presence of contaminant compounds and/or their influence on the background absorption spectrum against which

the water features are measured (Residuals, Baseline Shift, Slope Shift, Baseline Curvature, and $CH_4$; see description in Johnson et al. (2017)). Values for each metric were averaged across the injections for each sample and the equivalent value for a pure water sample run at the beginning of each run (EV) was subtracted from the sample value to obtain an anomaly value for the sample. Optimal models were selected to minimize the Bayes Information Criterion (BIC), with the caveat that highly collinear parameters were excluded (VIF function; Signorell, 2024). The stability and performance of the optimal

models were tested using iterative (n = 1000) split-sample training/testing in which 10% (7) random samples were withheld in the testing fold for each iteration. The models were then applied to the full dataset, and summaries of the modelled $\delta^2H$ and

$\delta^{18}O$ bias were generated with reference to taxonomic data obtained from the Global Biodiversity Information Facility database (Chamberlain and Boettiger, 2017; Chamberlain et al., 2025). All analyses were conducted in the R software environment (R Core Team, 2024) and all data and code are available on Zenodo (Bowen, 2025).

## 3 Results

$\delta^2H$ and $\delta^{18}O$ values for soil samples were similar for both analysis methods, with pervasive bias or extreme outlier values (Fig. 1A&B). The residual variance for the CRDS-IRMS comparison was somewhat higher than would be expected based solely on propagating the analytical uncertainties reported above, which is not unexpected for analysis of complex, real-world samples. The CRDS values for extracted plant waters, in contrast, exhibit a wider range of variation relative to the IRMS data and a tendency for large positive biases (Fig. 1C&D). CRDS $\delta^{18}O$ values of some plant samples, in particular, are as much as 19.6‰ higher than the IRMS values for the same samples. In total, 7 plant samples (13%) have $\delta^2H$ bias exceeding 8‰ and 28 (53%) have $\delta^{18}O$ bias greater than 1‰ – subjective thresholds which we use as representative of differences that would be interpreted as meaningful in most research studies. Most samples with large bias for one or both isotope systems were flagged as contaminated (indicated by red highlighting; 23/30) or suspect (yellow; 2/30) by the ChemCorrect software, suggesting that the CRDS analyses may have been biased by organic contaminants. The vendor's software also yielded many false positives, however, flagging 24 of 38 samples that did not exhibit large magnitude bias as contaminated (17) or suspect (7).

We found systematic relationships between the magnitude of the CRDS $\delta^2H$ and $\delta^{18}O$ bias and most of the spectral metrics. For $\delta^2H$ bias, the optimal model (lowest BIC) was a function of the product of the Slope Shift and $CH_4$ anomalies (Fig. 2A):

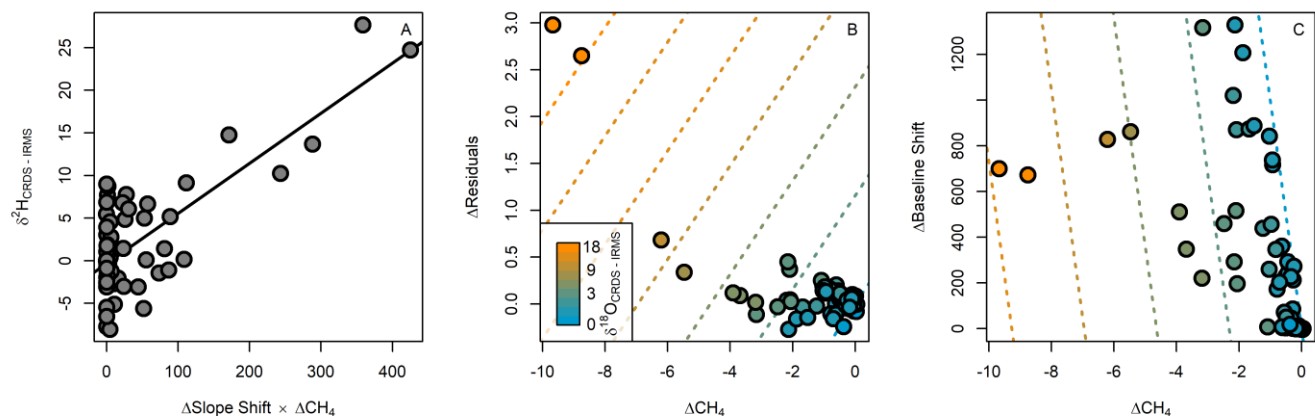

**Figure 2: Relationship between CRDS bias and spectral feature anomalies (relative to a pure water standard analysed in each CRDS run) included in the optimal models. A: $\delta^2H$ bias as a function of the product of baseline slope shift and the spectrally detected $CH_4$ concentration in the CRDS cavity. B: Partial response of $\delta^{18}O$ bias to $CH_4$ concentration and spectrum fitter residual value. C: Partial response of $\delta^{18}O$ bias to $CH_4$ concentration and spectral baseline shift. Lines in each panel show the fitted model response.**

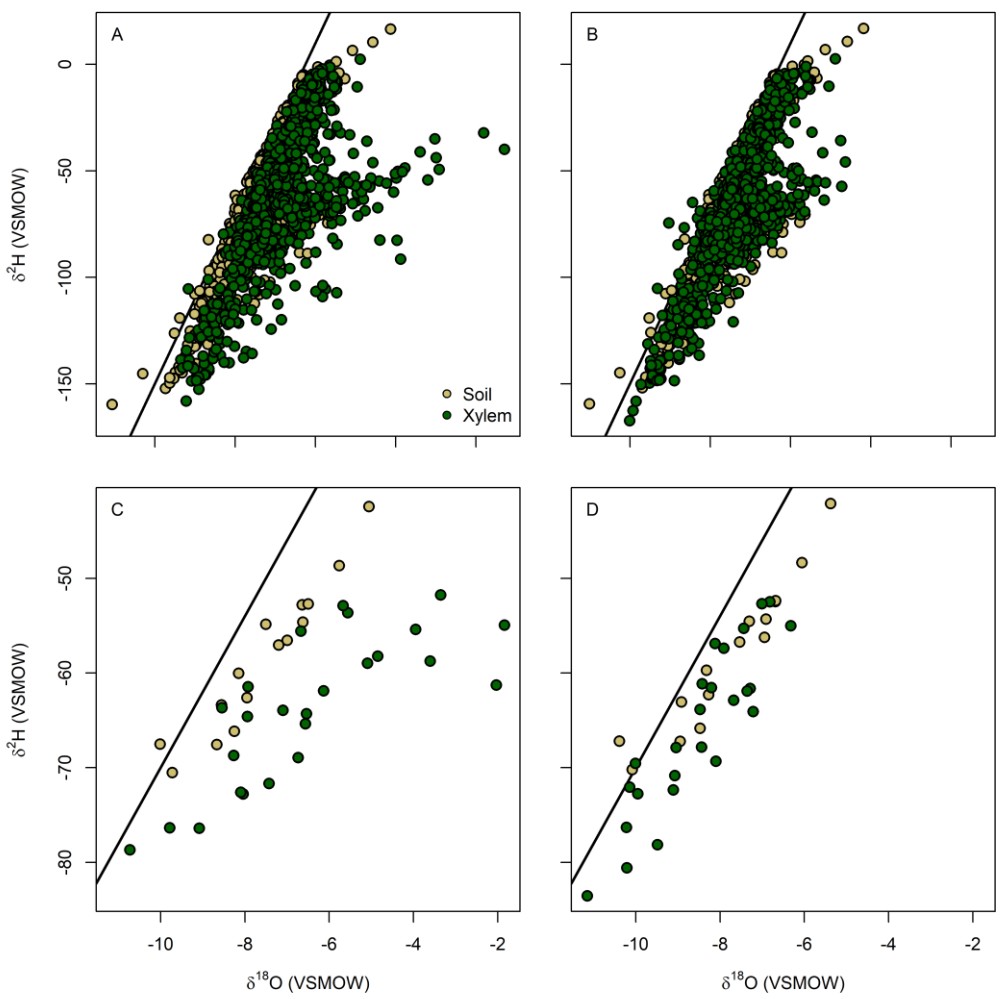

**Figure 3: CRDS data for water extracted from plant xylem and soil samples before (A and C) and after (B and D) model-based bias correction. Panels A and B show data from all sampling bouts at 12 U.S. National Ecological Observatory Network sites; panels C and D show data from a single bout (August 26, 2020) at the Harvard Forest site. The Global Meteoric Water Line ($\delta^2H = 8 \times \delta^{18}O + 10$) is shown in each panel.**

$$\delta^2 H_{CRDS-IRMS} = 0.0588 \times \Delta Slope\ Shift \times \Delta CH_4 - 0.32.$$

This model explained 58% of the variance in $\delta^2H$ bias, with a residual standard deviation of 4.1‰. For $\delta^{18}O$ bias, model selection including interactions between terms yielded an optimal model with four parameters that were highly collinear. As a

result, we opted to fit the $\delta^{18}O$ bias model without interactions, which gave an optimal model that was a linear combination of $CH_4$, Residual, and Baseline Shift anomalies (Fig. 2B&C):

$$\delta^{18}O_{CRDS-IRMS} = -1.295 \times \Delta CH_4 + 2.587 \times \Delta Residuals - 0.00130 \times \Delta Baseline\ Shift + 0.25.$$

The BIC and adjusted $R^2$ values for this model were only marginally different from those of the best model including interactions, and the variables in the model without interactions were not strongly collinear (variable inflation factors $\leq 5$). This model explained 99% of the variance in $\delta^{18}O$ bias, with a residual standard deviation of 0.36‰. The split-sample test showed that the optimal models were stable and performed well for out-of-sample prediction: the residual standard deviation for predictions made on test samples was 4.3‰ for $\delta^2H$ and 0.38‰ for $\delta^{18}O$, almost equalling the values for the full model.

We applied the optimal models to calculate bias corrections for the full plant and soil dataset. Approximately 33% of the plant samples (399 samples) and 1% of the soil samples (7) yielded $\delta^{18}O$ bias estimates > 1‰; only 5% of plant samples (62) and no soil samples had $\delta^2H$ bias estimates > 8‰. Our results showed strong but imperfect correspondence with ChemCorrect screening: the vast majority (94%) of samples with large bias estimates were flagged as contaminated or suspect, but false
positives may again be prevalent, with 34% of samples with modelled bias lower than 1 and 8‰ (for $\delta^{18}O$ and $\delta^2H$, respectively) being flagged by the vendor's software. Modelled bias estimates were as high as 76‰ for $\delta^2H$ and 33‰ for $\delta^{18}O$, and 15 (13) samples had $\delta^2H$ ($\delta^{18}O$) bias estimates that exceeded the maximum values in the data used to train the model.

Prior to bias correction, the dataset included many plant samples with isotopic values that fell well outside of the distribution
of the soil water data (which presumably represented the water sources used by many of the plants; Fig. 3A and C). After using the modelled values to bias-correct the data we found stronger correspondence between plant and soil data, with most plant sample values now falling within the envelope defined by the soils (Fig. 3B and D). Corrected data from an example sampling bout at one NEON site show a dramatic increase in overlap with potential soil water sources (Fig. 3C and D), with a small number of plant data showing low $\delta^2H$ and $\delta^{18}O$ values that might reflect uptake of unsampled deep soil water derived from
cool-season precipitation.

## 4 Discussion

Our results are consistent with other studies (e.g., Lazarus et al., 2016; Johnson et al., 2017; Herbstritt et al., 2024) in confirming that spectral bias is a persistent problem in CRDS isotope analysis of plant water samples but is rare for water extracted from soils. The data set reported here represents what is, to our knowledge, the most extensive and diverse assessment
of spectral bias in such measurements, and the bias values modelled using the CRDS spectral parameters show that the prevalence of bias varies dramatically across taxa (Table 1). High magnitude $\delta^{18}O$ bias is most frequent in woody taxa and appears to be particularly common in genera such as *Pinus*, *Quercus*, and *Artemisia*, although not all species are equally affected. Although many of these taxa are resinous or aromatic no single trait seems to unite them and others that show a

tendency for $\delta^{18}O$ bias. Strong $\delta^{18}O$ bias is uncommon in waters extracted from grass roots. High magnitude bias for $\delta^2H$ is much less common, and although it also shows strong taxonomic preference, this does not fully parallel that for $\delta^{18}O$ bias. Strong $\delta^2H$ bias is essentially absent among taxa which exhibit infrequent (20% of samples or fewer) $\delta^{18}O$ bias, but many species which commonly show bias for $\delta^{18}O$ exhibit none for $\delta^2H$.

Differences in the prevalence of spectral interference for different plant sample types have been demonstrated previously (e.g., Schultz et al., 2011; Nehemy et al., 2019; Herbstritt et al., 2024) and likely reflect differences in the composition and abundance of volatile organic compounds susceptible to extraction from these samples. Collectively, these results suggest that the potential for spectral interference to compromise isotope-based ecohydrological studies will vary markedly depending on the study system. Although researchers should exercise caution and consider conducting their own tests, the results shown here may help identify taxa with limited potential for spectral bias in measurements made with current-generation CRDS instruments (Picarro L2130-i and potentially L2140-i, which uses the same spectral absorption features when run in non-$^{17}O$ mode).

That said, our results also suggest that spectral interference bias in CRDS measurements may be correctable. We extended on the recent work of Herbstritt et al. (2024), who developed a set of correction equations using the $CH_4$ spectral metric, and show that the inclusion of other spectral metrics reported by the CRDS software can improve isotopic bias corrections (Fig. 2). Although methane has known interference in the wavelengths measured by the CRDS analysers, the utility of this metric for bias correction is most likely as a proxy for the presence of other interfering volatile organic compounds which are more common in plant tissue (Herbstritt et al., 2024). As such, it makes sense that other metrics which further describe changes in the shape of the absorption spectrum could provide additional information useful in detecting and correcting interference. The details of how these metrics are calculated are not publicly available from the instrument vendor and we can only speculate on their mechanistic connection to the observed isotopic data biases. They each describe deviations between the measured absorption spectrum and that expected for pure water and/or the factory-calibrated spectral baseline and their relationship to isotopic bias most likely reflects systematic patterns of distortion in the spectrum associated with common contaminant phases.

A common concern that has likely limited the use of *post-hoc* correction for CRDS spectral interference is that corrections may be application and/or instrument specific. Although we cannot confidently argue that the correction approach developed here will be globally applicable, we note that the same correction equations appear, based on direct (Fig. 1) and indirect evidence (Fig. 3), to successfully correct bias for vacuum-extracted water from a large and diverse range of plants and soils. Given the inevitable wide variation in VOC composition and concentration among these samples this result suggests that a single correction framework may be applicable across most ecohydrological applications and systems. It is more difficult to judge whether the model parameter values calibrated here will be applicable to other analysers given variation in instrument optics and calibrations, but we found that a single model calibration could successfully be applied to data generated on

two different L2130-i analysers: the difference in the mean model residuals for samples run on the two instruments was small relative to the dispersion of the residuals (0‰ for $\delta^2H$ and 0.16‰ for $\delta^{18}O$) and not significantly different from zero (t-test, p = 0.99 for $\delta^2H$; Wilcoxon rank sum test, p = 0.09 for $\delta^{18}O$), suggesting that the same optimal models accurately corrected bias on both instruments. That said, the coefficient describing $CH_4$ sensitivity of $\delta^{18}O$ bias on our analysers is similar but not identical to that fit by Herbstritt et al. (2024), suggesting that some variability may exist even between analysers of the same model. Further testing and comparative calibration of bias-correction algorithms is thus warranted.

## 5 Conclusion

Our survey of > 1800 samples shows that spectral bias is prevalent in CRDS $\delta^2H$ and $\delta^{18}O$ measurements of water extracted from plant tissues, that this bias varies substantially between plant types, and that soil-extracted waters are largely immune from bias. We also show that robust bias-correction algorithms can be developed using commonly reported spectral metrics and applied successfully across sample types on data from two different analysers. Although we advocate for further testing and comparison across laboratories, this work supports results from other groups (Schultz et al., 2011; Herbstritt et al., 2024) in suggesting that spectroscopic measurement, combined with *post-hoc* bias correction, may be a robust, effective, and efficient method for isotope ratio analysis of water samples in ecohydrology and related fields.

### Code and data availability

All data and code used to analyse the data and create the figures is archived on Zenodo (Bowen, 2025).

### Author contribution

GJB: Conceptualization, Data curation, Formal analysis, Funding acquisition, Methodology, Software, Writing – original draft preparation. SB: Investigation, Data curation, Writing – review & editing. SC: Investigation, Data curation, Writing – review & editing.

**Table 1: Taxonomic composition and prevalence of high-magnitude bias, as modelled based on spectral features, in the plant dataset. Taxa marked with * are also represented in the CRDS-IRMS model calibration dataset.**

| Taxon | Vernacular | Count | Modelled bias $\delta^2H > 8$ | $\delta^{18}O > 1$ |
|---|---|---|---|---|
| *Geum rossii* | Ross's avens | 15 | 33% | 100% |
| *Pinus contorta* | Lodgepole pine | 19 | 0% | 79% |
| *Diosypros virginiana* | Persimmon | 15 | 0% | 73% |
| *Medicago sativa** | Alfalfa | 29 | 69% | 72% |
| *Pinus palustris* | Longleaf pine | 35 | 0% | 71% |
| *Quercus falcata** | Spanish oak | 20 | 0% | 70% |
| *Artemisia frigida* | Fringed sage | 29 | 21% | 69% |
| *Pinus sabiniana** | Gray pine | 6 | 0% | 67% |
| *Liquidambar styraciflua** | Alligator wood | 20 | 0% | 65% |
| *Pinus strobus** | Weymouth pine | 30 | 7% | 63% |
| *Juglans nigra* | Black walnut | 30 | 13% | 60% |
| *Betula alleghaniensis* | Yellow birch | 30 | 3% | 60% |
| *Artemisia absinthium** | Wormwood | 28 | 32% | 57% |
| *Pseudotsuga menziesii* | Douglas fir | 34 | 0% | 50% |
| *Artemisia tridentata* | Big sagebrush | 19 | 11% | 47% |
| *Quercus laevis* | Catesby's oak | 15 | 0% | 47% |
| *Picea mariana** | Black spruce | 33 | 0% | 42% |
| *Liriodendron tulipifera** | Tulip poplar | 30 | 7% | 40% |
| *Gaultheria shallon* | Shallon | 15 | 0% | 40% |
| *Quercus wislizeni** | Interior live oak | 10 | 0% | 40% |
| *Quercus douglasii** | Blue oak | 8 | 13% | 38% |
| *Quercus rubra** | Red oak | 60 | 7% | 37% |
| *Minuartia obtusiloba* | Alpine stitchwort | 15 | 0% | 33% |
| *Quercus stellata** | Post oak | 15 | 0% | 33% |
| *Vaccinium arboreum** | Farkleberry | 20 | 0% | 30% |
| *Bromus hordaceus** | Soft brome | 7 | 14% | 29% |
| *Abies balsamea** | Balsam Fir | 28 | 0% | 29% |
| *Tsuga canadensis* | Black hemlock | 30 | 0% | 27% |
| *Acer rubrum** | Red maple | 60 | 5% | 25% |
| *Quercus alba** | Stave oak | 28 | 4% | 25% |
| *Acer saccharum** | Sugar maple | 30 | 0% | 23% |
| *Abies lesiocarpa* | Alpine fir | 15 | 0% | 20% |
| *Sorghastrum nutans** | Indiangrass | 15 | 0% | 20% |
| *Bouteloua curtipendula* | Sideoats grama | 14 | 0% | 14% |
| *Aristida beyrichiana* | Wiregrass | 15 | 0% | 13% |
| *Salix sp.* | Willow | 15 | 0% | 13% |
| *Tsuga heterophylia* | Western hemlock | 15 | 0% | 13% |
| *Poa pratensis** | Kentucky bluegrass | 28 | 4% | 11% |
| *Bromus inermis** | Hungarian brome | 29 | 0% | 10% |
| *Quercus germinata* | Sand live oak | 15 | 0% | 7% |
| *Schizachyrium scoparium** | Little bluestem | 43 | 0% | 7% |
| *Taxus brevifolia* | Pacific yew | 15 | 0% | 7% |
| *Betula papyrifera** | Paper birch | 31 | 0% | 3% |
| *Eriogonum effusum* | Spreading buckwheat | 29 | 0% | 3% |
| *Bouteloua gracilis* | Blue grama | 19 | 0% | 0% |
| *Carex rupestris* | Curly sedge | 15 | 0% | 0% |
| *Elymus elymoides* | Bottlebrush squirreltail | 15 | 0% | 0% |
| *Festuca spp.* | Fescue | 19 | 0% | 0% |
| *Hesperostipa comata* | Needle-and-thread grass | 17 | 0% | 0% |
| *Poaceae sp.* | Grass | 19 | 0% | 0% |
| *Smilax bona-nox** | Catbrier | 15 | 0% | 0% |
| *Thuja plicata* | Pacific red cedar | 15 | 0% | 0% |

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
