# Peer review of "Technical Note: Spectral correction for cavity ringdown isotope analysis of plant and soil waters"

_EGUsphere, 2025_

## Author Response (AR1)

**Reviewer 1**

**General comments:**

In the manuscript by Bowen et al. the authors present a post-correction method to correct for the spectral bias in isotope data of laser-based (CRDS) isotope analyzers. Two instruments of the same type were used to analyze cryogenically extracted soil and plant water from around 1200 plant and 700 soil samples. For a subset, IRMS data were also available and assumed to represent the 'true' values. The authors applied and optimized models for the two isotopes separately where the isotopic bias is described as a function of 5 relevant metrics, reflecting potential contamination. The R-codes are available on zenodo.

The spectral interference problem was already described 15 years ago, but correction schemes are still not provided by the manufacturers. Removal of contaminants or flagging of suspicious data are the main focus. Several individual post-correction approaches have been published and some of them might be broadly useful.

Against this, in this study several relevant metrics are used for optimal models and beyond this, applied to a large and diverse dataset. Therefore, this work will highly be appreciated by the community.

In general, the manuscript is very well written, well structured and easy to follow.

I therefore recommend publication with only some minor revisions.

WE APPRECIATE THE POSTIVITVE FEEDBACK. WE HAVE MADE MINOR CHANGES AND ADDITIONS IN RESPONSE TO THE CONSTRUCTIVE FEEDBACK, AS DESCRIBED BELOW, WHICH WE BELIEVE ADDRESS THE REVIEWER'S INPUT AND IMPROVE THE MANUSCRIPT.

**Specific Comments:**

L 38 and throughout the manuscript: please check the order of the references (oldest – newest)

UPDATED AS REQUESTED

Figure 1: The symbols for "corrected" are quite large and probably overlap a lot of symbols of the "raw" data. I suggest reducing the symbolsize

IT IS INEVITABLE THAT THERE WILL BE SOME OVERLAP…SOME SAMPLES HAVE LITTLE TO NO CORRECTION APPLIED…BUT WE HAVE REDUCED THE SIZE OF THE SYMBOLS SOMEWHAT AND ADDED A NOTE TO THE CAPTION (AS SUGGESTED BY REVIEWER 2) TO HELP THE READER INTERPRET THE FIGURE.

L 119: Wouldn't it be impressive to show a few selected plant species where the correction was highly effective in a separate figure?

WE HAVE ADOPTED THIS SUGGESTION AND NOW INLCUDE TWO ADDITIONAL PANELS IN FIGURE 3 WHICH HIGHLIGHT RESULTS FROM A SINGLE SAMPLING BOUT THAT CLEARLY SHOW THE IMPROVEMENT IN THE CONSISTENCY OF PLANT AND SOIL SOURCE WATER DATA. WE THINK THIS IS MORE EFFECTIVE THAN TRYING TO SUMMARIZE OR ILLUSTRATE THE EFFECTIVENESS OF THE CORRECTION FOR PARTICULAR PLANT SPECIES BECAUSE THE SOURCE WATER VALUES MAY BE HIGHLY VARIABLE ACROSS SITES AND/OR SAMPLING BOUTS FOR DIFFERENT INDIVIDUALS OF THE SAME SPECIES.

**Technical corrections:**

L 203: There is something missing in the table caption after "...spectral"

FIXED

**Reviewer 2**

**General comments:**

The manuscript from Bowen et al. addresses a much-discussed topic, namely the systematic measurement differences between isotope ratio mass spectrometry (IR-MS) and cavity ring-down spectroscopy (CRDS) of cryogenically extracted water samples from plants and soil and addresses the question of what causes them.

The authors compared the results of isotope measurements, using IR-MS and CRDS of 16 soil samples and 54 plant samples. The observed discrepancies were attributed to spectral interferences in the CRDS measurements, which are less relevant in IR-MS measurements. Therefore, the MS results were considered the "true" values. The authors used spectral parameters from the CRDS measurements to create multivariate models and to correct for these interferences. The corrections were then applied to a large CRDS dataset of plant and soil samples and the plants were also differentiated by different plant taxa. The manuscript shows that especially for plant samples correction algorithms have to be found to obtain reliable CRDS data and to correct spectral interferences.

Overall, the manuscript is well-structured and clearly written. I believe it only requires minor revisions and specification of a few details.

WE APPRECIATE THE POSTIVITVE FEEDBACK. WE HAVE MADE MINOR CHANGES AND ADDITIONS IN RESPONSE TO THE CONSTRUCTIVE FEEDBACK, AS DESCRIBED BELOW, WHICH WE BELIEVE ADDRESS THE REVIEWER'S INPUT AND IMPROVE THE MANUSCRIPT.

**Specific comments:**

L66: under which conditions where the samples stored? (e.g., temperature ...)

THE SAMPLES WERE STORED IN THEIR SEALED VIALS AT ROOM TEMPERATURE. WE HAVE ADDED THIS DETAIL TO THE METHODS SECTION.

L78: no sample was classified as contaminated using ChemCorrect software, yet the measurement differences between MS and CRDS were attributed to spectral interferences. For me the question arises why none of the measured samples were classified as contaminated. Many other working groups use the ChemCorrect software to classify samples and to make a possible correction for contaminated samples or to exclude these samples. Where the corrections also tested for as contaminated flagged samples?

WE HAVE REANALYZED ALL OF OUR DATA FILES WITH CHEMCORRECT AND NOW REPORT UPDATED RESULTS WHICH DEMONSTRATE: 1) STRONG ASSOCIATION BETWEEN THE MEASURED AND MODELED BIAS ESTIMATES FROM OUR WORK AND CONTAMINANT FLAGGING BY THE CHEMCORRECT SOFTWARE, AND 2) A TENDENCY FOR THE VENDOR'S SOFTWARE TO FLAG MANY SAMPLES WHERE LITTLE OR NO BIAS WAS OBSERVED OR ESTIAMTED IN OUR WORK. THESE FINDINGS HAVE BEEN ADDED TO THE RESULTS SECTION AND, AS THE REVIEWER SUGGESTS, SHOULD BE QUITE USEFUL TO OTHER LABORATORIES THAT ARE USING THE COMMERCIAL SOFTWARE.

Q 81: Were the samples analyzed with MS treated in the same way as the samples measured with CRDS, i.e. was activated carbon also added ore just to the CRDS-samples?

YES, THE TREATMENT WAS CONDUCTED ON THE BULK EXTRACTED WATER PRIOR TO TAKING ALIQUOTS FOR CRDS AND IRMS ANALYSIS. WE NOW STATE THIS IN THE METHODS SECTION: "IRMS ANALYSES WERE CONDUCTED ON RESIDUAL WATER FROM THE CRDS ANALYSIS VIALS."

L 86: Please clarify what is meant by "several months" — how many months exactly?

WE WERE PURPOSEFULLY INSPECIFIC HERE BECAUSE, GIVEN THE NUMBER OF SAMPLES, THE CRDS ANALYSES OCCURRED OVER A ~2 YEAR PERIOD. THE TIME LAG BETWEEN CRDS AND IRMS ANALYSIS THUS VARIED FOR DIFFERENT SAMPLES, RANGING FROM 5 TO 9 MONTHS. WE HAVE ADDED THIS INFORMATION ("...STORED FOR UP TO 9 MONTHS...") TO THE METHODS.

L88: A criterion for identifying samples with too high deuterium excess values is mentioned, but it is not defined. How was the criterion defined for too high deuterium excess value?

WE HAVE ELABORATED THE SCREENING CRITERIA IN THE FINAL TWO SENTENCES OF THIS PARAGRAPH

L183: I would also like to point out, that larger differences are likely, especially when using different generations of analyzers. Therefore, I am concerned that the presented correction models might not be transferable across instruments (as mentioned in the manuscript). It would be interesting to see and test these corrections for other devices (not in this manuscript, rather for future measurements).

WE AGREE, HENCE THE CAUTIONARY STATEMENT IN THE TEXT. WE ARE OPTIMISTIC THAT THE CORRECTION APPROACH WILL BE GENERALLY APPLICABLE, AT LEAST WITH THIS GENERATION OF PICARRO ANALYZERS, BUT ADDITIONAL TESTING BY OTHER LABS WILL BE NEEDED TO CONFIRM THIS AND ESTABLISH THE DEGREE TO WHICH CORRECTION MODEL PARAMETERS VARY BETWEEN INSTRUMENTS.

Figure 1: the legend is misleading. It would be better to change the caption to: "raw soil data", "raw xylem data", "corrected soil data" and "corrected xylem data" instead of "raw", "soil", "xylem", "corrected".

WE HAVE REVISED THE LEGEND AS SUGGESTED.

Figure 1: Do the raw soil data in Figure 1 A correspond to the corrected soil data and if so, are the raw soil data not visible in the diagram because they overlap with the corrected data? This could be clarified either in the caption or main text.

YES, THEY DO. WE HAVE ADDED A STATEMENT TO THE CAPTION CLARIFYING THIS.

REFERENCES:

Chang, E., Wolf, A., Gerlein-Safdi, C., & Caylor, K. K. (2016). Improved removal of volatile organic compounds for laser-based spectroscopy of water isotopes. Rapid Communications in Mass Spectrometry, 30(6), 784-790. https://doi.org/https://doi.org/10.1002/rcm.7497

---

## Author Response (AR2)

**Associate Editor**

Thank you very much for your manuscript revisions. I have a few suggestions for changes: Could you please change the color or make the symbols of the "plant samples" hollow in Fig. 3 (new), so that the soil samples become visible (as currently they are overlapped by the plant samples).

WE HAVE EXPERIMENTED WITH A VARIETY OF DIFFERENT SYMBOL TYPES. EVEN WITH OPEN OR SEMI-TRANSPARENT SYMBOLS, THE VAST NUMBER OF XYLEM DATA POINTS CAUSES MOST OF THE DISTRIBUTION OF SOIL DATA TO BE OBSCURED. THE REVISED FIGURE SUBMITTED HERE REPRESENTS THE BEST COMPROMISE WE WERE ABLE TO FIND – WE HAVE ADPOTED A SMALLER AND SIMPLER SYMBOL FOR THE XYLEM DATA IN THE TOP TWO PANELS, WHICH ALLOWS THE READER TO AT LEAST SEE THE RANGE OF DISTRIBUTION OF SOIL VALUES.

In my opinion, it would be a great benefit to show the "failure" of the ChemCorrect software (e.g. in an Appendix figure or table) as so many colleagues in our community rely on this flagging system assuming the data is "fine" after conducting the analysis.

WE HAVE ADDED A SUPPLEMENTAL FIGURE THAT WE HOPE ADDRESSES THIS SUGGESTION – IT SHOWS THE DISTRIBUTION OF OBSERVED AND MODELLED BIASES FOR EACH CHEMCORRECT CONTAMINATION FLAG VALUE (GREEN/YELLOW/RED).

Could you also please add the R2 values for the fitted models in Fig. 2.

DONE

Would it be possible to analyze a "species-effect" for a subset of your plant samples to underline your arguments in the discussion or at least show the variability across different species?

ALTHOUGH WE ARE NOT ENTIRELY SURE WE UNDERSTAND THE SUGGESTION, WE HAVE RESPONDED BASED ON OUR INTERPRETATION OF THE INTENT. WE HAVE PROVIDED MORE INFORMATION ON THE TAXON-SPECIFIC BIASES BY EXPANDING THE INFORMATION PRESENTED IN TABLE 1 OF THE MANSCRIPT. THIS TABLE NOW SHOWS THE NUBER OF SAMPLES, AVERAGE AND STANDARD DEVIATION OF THE MODELLED BIASES, AND PREVELANCE OF HIGH-MAGNITUDE BIAS FOR EACH SPECIES IN THE DATA SET.

Could you please add 1-2 sentences to your revised manuscript addressing this "L183_ instrument specific-corrections…" reviewer comment?

WE COULD HAVE BEEN MORE DIRECT IN OUR REPLY TO THIS REVIEWER COMMENT…THE MANUSCRIPT ALREADY INCLUDES A FULL PARAGRAPH (LINES 184-197) THAT RAISES AND DISCUSSES THIS CONCERN, WHICH WE COPY BELOW FOR REFERENCE. WE BELIEVE THAT THIS ACCURATELY AND COMPREHENSIVELY REPRESENTS WHAT WE ARE ABLE TO SAY ABOUT THE SUBJECT AT THIS POINT, SO WE HAVE NOT MADE ADDITIONAL MODIFICATIONS.

"A common concern that has likely limited the use of *post-hoc* correction for CRDS spectral interference is that corrections may be application and/or instrument specific. Although we cannot confidently argue that the correction approach developed here will be globally applicable, we note that the same correction equations appear, based on direct (Fig. 1) and indirect evidence (Fig. 3), to successfully correct bias for vacuum-extracted water from a large and diverse range of plants and soils. Given the inevitable wide variation in VOC composition and concentration among these samples this result suggests that a single correction framework may be applicable across most ecohydrological applications and systems. It is more difficult to judge whether the model parameter values calibrated here will be applicable to other analysers given variation in instrument optics and calibrations, but we found that a single model calibration could successfully be applied to data generated on two different L2130-i analysers: the difference in the mean model residuals for samples run on the two instruments was small relative to the dispersion of the residuals (0‰ for $\delta^2$H and 0.16‰ for $\delta^{18}$O) and not significantly different from zero (t-test, p = 0.99 for $\delta^2$H; Wilcoxon rank sum test, p = 0.09 for $\delta^{18}$O), suggesting that the same optimal models accurately corrected bias on both instruments. That said, the coefficient describing $CH_4$ sensitivity of $\delta^{18}$O bias on our analysers is similar but not identical to that fit by Herbstritt et al. (2024), suggesting that some variability may exist even between analysers of the same model. Further testing and comparative calibration of bias-correction algorithms is thus warranted."